# Proline-based solution maintains cell viability and stemness of canine adipose-derived mesenchymal stem cells after hypothermic storage

Pongsatorn Horcharoensuk[1], Sunantha Yang-en[1], Amarin Narkwichean[2], Ruttachuk Rungsiwiwut[1]*

1 Department of Anatomy, Faculty of Medicine, Srinakharinwirot University, Bangkok, Thailand,
2 Department of Obstetrics and Gynaecology, Faculty of Medicine, Srinakharinwirot University, Nakhon Nayok, Thailand

* ruttachuk@swu.ac.th

**Data Availability Statement:** All relevant data are within the paper and its Supporting information files.

## Abstract

Transportation of mesenchymal stem cells (MSCs) under hypothermic conditions in 0.9% normal saline solution (NSS) might increase cell death and alter the stemness of MSCs. The present study aimed to evaluate the effect of proline-based solution (PL-BS) on cell viability and the stemness of newly established canine adipose-derived mesenchymal stem cells (cAD-MSCs) under hypothermic conditions. Characterized cAD-MSCs were stored in 1, 10, and 100 mM PL-BS or NSS at 4°C for 6, 9, and 12 hours prior to an evaluation. The results demonstrated that storage in 1 mM PL-BS for 6 hours decreased cell apoptosis and proliferation ability, but improved cell viability and mitochondrial membrane potential. cAD-MSCs maintained their high expression of CD44 and CD90, but had a low expression of CD34 and MHC class II. Trilineage differentiation ability of cAD-MSCs was not affected by storage in 1 mM PL-BS. Gene expression analysis demonstrated that immunomodulatory genes, including IDO, HGF, PGE-2, and IL-6, were upregulated in cAD-MSCs stored in 1 mM PL-BS. In conclusion, PL-BS can be effectively applied for storing cAD-MSCs under hypothermic conditions. These findings provide a new solution for effective handling of cAD-MSCs which might be promising for clinical applications.

## Introduction

Mesenchymal stem cells (MSCs) are multipotent cells which are able to regenerate themselves and differentiate into many cell types [1]. The therapeutic potential of MSCs, especially in regenerative medicine, is widely recognized due to their stemness [2], including capacity to migrate to injured sites, immunomodulation, anti-inflammation, anti-apoptosis, neo-angiogenesis, antimicrobial activity, and tissue regeneration. Recently, clinical trials of MSC therapies have been performed to treat several diseases, such as cardiovascular, neurological,

**Funding:** Faculty of Medicine, Srinakharinwirot University (MED-GRAD-150; 511/2563 and MED-RES-200; 217/2563). The funders had no role in study design, data collection and analysis, decision to publish, or preparation of the manuscript.

**Competing interests:** The authors have declared that no competing interests exist.

immune-mediated, and traumatic-related diseases, and so forth. The vast majority of the clinical trials have been in phase 2 (61.0%), while others have been in phases 1 (30.8%), 3 (7.5%), and 4 (0.7%), respectively. In addition, those trials have been conducted in 51 countries globally, especially in China (25.25%), United States of America (20.6%), and Spain (7.67%) [3]. Based on cell therapy, the workflow consists of three major complicated processes. The first is an extraction of MSCs from donors; it is required that this step is performed at the cell processing center. The most common sources of MSCs in adult humans are bone marrow, adipose tissues, and peripheral blood, respectively [4]. Meanwhile, those in companion animals include adipose tissues and bone marrow. In addition, adipose tissues can be easily and practically harvested from domesticated dogs and cats by a routine sterilization procedure [5]. Second, it is required that MSCs are isolated, cultured, multiplied, and preserved under freezing conditions for long-term storage. Third, the cells have to be defrosted and subsequently cultured prior to transplantation. Finally, the collected and processed cells require transport solutions for shipment before they are applied to the patients [6].

The transportation time of stem cell packages varies depending on the distance between the cell processing center and the hospital. In general, short-term transportation requires temperature control, either at a severely hypothermic state (2–8˚C) or at room temperature (25˚C). On the contrary, preservation for long-term transportation needs a frozen state (below 0˚C) so that the cryoprotectant works to prevent cell death [7, 8]. Previously, effects of storage temperature on MSCs have been widely studied. Gálvez-Martín and colleagues, who were studying the influence of storage temperature on human adipose-derived stem cells (hADSCs) kept in different storage solutions at 4˚C, 25˚C, and 37˚C for up to 60 hours, demonstrated that hADSCs are able to maintain the highest viability at 4˚C, when compared to viability at 25˚C and 37˚C, respectively [9]. Gniadek and colleagues demonstrated that peripheral blood-derived mononuclear cells (MNCs) preserved in 5% human serum albumin solution and Hypothermosol FRS at 4˚C possess higher cell viability than at 25˚C [10]. These results emphasized the importance of suitable solutions for maintaining stem cell viability when stored at 4˚C. For short-term storage and cell transportation, a number of solutions can be applied as the hypothermic solution, such as 0.9% normal saline solution (NSS), Ringer solution, Plasma-Lyte A, phosphate-buffered saline (PBS), platelet-rich plasma, etc. In human clinical trials, 0.9% NSS has been used for short-term transportation of human MSCs [11]. However, negative effects of NSS on cell viability and MSC stemness have been reported. For instance, Chen and colleagues demonstrated that storing human umbilical cord mesenchymal stem cells (hUC-MSCs) in NSS at 4˚C significantly decreases cell viability and adhesion capacity in a time-dependent manner but increases the population doubling time. It can be implied that hUC-MSCs cannot be recovered efficiently after being stored in hypothermic conditions and may lose their proliferation ability after transplantation into the patients. Sohn and colleagues reported that storing human bone marrow stem cells (hBMSCs) in NSS at 4˚C decreases their viability, level of colony-forming units, and trilineage differentiation ability. This suggests that short-term preservation of MSCs in NSS at 4˚C may not be an effective method as a decrease in cell proliferation and differentiation capacity occurs [12, 13].

Therefore, the objective of the present study was to improve the efficiency of NSS as a solution for short-term storage, under hypothermic conditions, of canine adipose-derived mesenchymal stem cells (cAD-MSCs). To date, there have been no reports about using proline, a novel cryoprotective agent, as a supplement for cell transportation solutions. Thus, proline would be a great candidate for studying protective effects during hypothermic conditions.

## Materials and methods

### Animal use

The protocols of animal use in the present study were approved by the Institutional Ethics Committee of Srinakharinwirot University (COA/AE-017-2563). Subcutaneous or periovarian fat samples were collected during routine castration or ovariohysterectomy procedures from three dogs at iVET animal hospital. The inclusion criteria for the dogs were as follows: (1) age between one and three years, (2) complete vaccination based on AAHA guidelines, (3) ecto-parasite prevention three months prior to surgery, and (4) a normal blood profile.

### Experimental designs

In the present study, samples were completely randomized and divided into two parts, which were (1) isolation and characterization of cAD-MSCs and (2) evaluation of the effect of proline-based solution (PL-BS) on cell viability, mitochondrial membrane potential (MMP), proliferation ability, specific surface markers, trilineage differentiation, and immunomodulatory properties of cAD-MSCs after storage in a hypothermic state.

In the first part, cAD-MSCs were isolated from fat tissues of the dogs and evaluated for stemness. In the second part, the cAD-MSCs were divided into three groups: non-storage control, hypothermic-storage in NSS, and hypothermic-storage in proline-based solution (PL-BS). L-proline (Catalog number P0380, Sigma-Aldrich®, USA) was prepared in NSS at concentrations of 1, 10, and 100 mM. The harvested cAD-MSCs in groups 2 and 3 were stored at a concentration of $1x10^6$ cells per 1 ml of storage solution and kept at 4˚C for 6, 9, and 12 hours in order to mimic the hypothermic transportation conditions. cAD-MSCs growing at 37˚C, 5% $CO_2$, were used as the non-storage control group.

### Isolation and expansion of canine adipose-derived mesenchymal stem cells

Adipose tissues were isolated from two male dogs and one female dog, kept in sterile containers filled with sterile NSS, and transferred to the stem cell laboratory within two hours. Fat tissues were washed in 70% ethanol, followed by PBS. The vessels were removed, and the tissues were minced into small pieces (<1 mm.) before tissue digestion was performed by adding 0.5 mg/ml collagenase type IV in PBS (Catalog number 17104019, Gibco, Thermo Fisher Scientific, USA) and incubating for 1 hour at 37˚C, 5% $CO_2$. After that, the enzyme activity was neutralized with a culture medium containing 10% fetal bovine serum (FBS, Catalog number S1810, Biowest, USA). Afterwards, the cell suspension was filtered through a 40 μm cell strainer (Catalog number 93040, SPL Life Sciences, South Korea) and centrifuged at 2,300 rpm for 5 minutes. Finally, the supernatant was discarded and cell pellets were resuspended in the culture medium consisting of MEM alpha modification (Catalog number SH30265.02, Hyclone, USA), 10%FBS, 1% penicillin/streptomycin (Catalog number 03-031-1B, Biological Industries, Israel), and 10 ng/ml human fibroblast growth factor basic recombinant protein (FGF-2/bFGF, Catalog number PHG0026, Invitrogen, USA). The cell suspension was seeded onto a culture plate and incubated at 37˚C, 5% $CO_2$, and the culture medium was replaced every other day. These cells were designated as passage 0 (P0). The expansion procedure was performed when the cells reached 80–90% confluence. In brief, the culture medium was discarded and the adherent cells were washed with PBS to remove the remaining residues. Cell dissociation was performed using 0.05% trypsin/EDTA (Catalog number 03-051-5B, Biological Industries, Israel) and incubating at 37˚C, 5% $CO_2$ for 5 minutes. After incubation, the cells were gently dispersed using pressure from a pipette, and enzyme activity was neutralized by adding a culture medium containing 10% FBS. The cell suspension was centrifuged at 2,300

rpm for 5 minutes, and subsequently, the supernatant was discarded and the cell pellet was resuspended with the culture medium. Eventually, the cell suspension was seeded onto the culture plate, and these cells were designated as passage 1 (P1). The expansion protocol was repeated to multiply the cell number. Cells from passages 3 to 5 (P3-P5) were used in the experiments.

## Cell viability and apoptotic/necrotic fraction assay

Both the non-stored and stored cAD-MSCs were evaluated for the percentage of live cells and the apoptotic/necrotic cell fraction using ApoScreen® Annexin V Apoptosis Kit-FITC (Catalog number 10010–02, SouthernBiotech, USA). In brief, the cells were collected, washed with PBS, and resuspended in a 1x binding buffer. Annexin V-FITC antibody was added into the cell suspension, which was then incubated at 4°C for 15 minutes; after that, propidium iodide (PI) was added. Live and apoptotic/necrotic cells were gated to $1x10^4$ cells and then analyzed by Guava easyCyte 5HT Flow Cytometer (Millipore, USA).

## Mitochondrial membrane potential assay

The cAD-MSCs were evaluated for MMP using the TMRE-Mitochondrial Membrane Potential Assay Kit (Catalog number ab113852, Abcam, UK). In brief, 250 nM TMRE was added into $1x10^5$ cells/well of both non-stored and stored cAD-MSCs in a suspension condition; the cell suspension was incubated for 15 minutes and then washed twice with 0.2% BSA in PBS prior to being loaded into a 96 well-plate and evaluated for fluorescence intensity using a microplate reader (Synergy HT, BIO-TEK, USA) with Ex/Em = 549/575 nm.

## Proliferation assay

The non-stored and stored cAD-MSCs were collected, washed, and seeded back into 96 well-plates at a concentration of $1x10^4$ cells/well, and incubated with 5%$CO_2$, at 37°C for 24 hours. Cell proliferation was evaluated using Cell Proliferation Kit I (MTT, Catalog number 11465007001, Roche Diagnostics, Germany). In brief, 10 μl of MTT solution I was added to the cell culture plate and incubated for 4 hours. Therefore, 100 μl of MTT solution II was added to the cell culture plate and cultured for 24 hours. The absorbance was measured by using a microplate reader (HiPo MPP-96 Microplate Photometer, Biosan, Latvia) at 550 nm.

## Expression of specific surface antigens

An analysis of specific surface antigens for MSCs, including CD44, CD90, CD34, and MHC class II, was performed by flow cytometry. In brief, the cell pellets of non-stored and stored cAD-MSCs were washed with PBS and resuspended in cold flow buffer (PBS with 5% bovine serum albumin (BSA), Catalog number 03-010-1B, Biological Industries, Israel) at a concentration of $5x 10^5$ cells/tube stained with conjugated antibody, for 30 minutes, and kept at 4°C with light protection. After incubation, the cell suspension was washed once with a cold flow buffer and resuspended with 400 μl of cold flow buffer. Finally, the cell surface antigens were gated to $1x10^4$ cells and then analyzed by flow cytometry. The antibodies used in the present study were anti-CD44-FITC Monoclonal Antibody (Catalog number 11-5440-42, Invitrogen, Thermo Scientific, USA), anti-MHC class II-FITC (Catalog number 11-5909-42, Invitrogen), anti-CD34-PE (Catalog number MA1-81855, Invitrogen), and anti-CD90-PE (Catalog number 12-5900-42, Invitrogen).

## Trilineage differentiation assay

The non-stored and stored cAD-MSCs were both induced for differentiation into three lineages, which were adipogenic, chondrogenic, and osteogenic lineages, in order to evaluate the differentiation potential. Adipogenic differentiation was adapted from previous studies [14, 15] and supplemented by Chemicon® Mesenchymal Adipogenesis Kit (Catalog number SCR020, Sigma-Aldrich®, USA). Adipogenic induction medium consisted of high-glucose DMEM, 15% rabbit serum, 1 μM dexamethasone, 10 μg/ml insulin, 200 μM indomethacin, 0.5 mM isobutyl-methyl-xanthine (IBMX), and 1% penicillin/streptomycin. The medium was changed every 1–2 days. On day 21, cells were fixed with 4% paraformaldehyde (PFA), washed with PBS, and stained with oil red O solution. ChondroMAX differentiation medium (Catalog number SCM123, Sigma-Aldrich®, USA) was used for chondrogenic differentiation. The chondrogenic differentiation medium was changed every 1–2 days. On day 21, cells were fixed with 4% PFA, stained, and washed with distilled water containing alcian blue solution (Catalog number B8438, Sigma-Aldrich®, USA). StemPro™ Osteogenesis Differentiation Kit (Catalog number A1007201, Gibco, USA) was used for osteogenic differentiation. The osteogenic differentiation medium was changed every 3–4 days. On day 21, cells were fixed with 4% PFA, washed with distilled water, and stained with alizarin red S solution (Catalog number A5533, Sigma-Aldrich®, USA).

## Immunomodulatory gene expression

Total RNA was extracted from non-stored and stored cAD-MSCs by using the GeneJET RNA Purification Kit (Catalog number K0731, Thermo Scientific, USA) and following the manufacturer's protocol. Total RNA concentration was quantified by Nanodrop 2000c spectrophotometer (Thermo Scientific, USA). Total RNA was reversely transcribed for cDNA synthesis by using the Superscript III Enzyme Reverse Transcriptase kit (Catalog number 18080093, Invitrogen, Thermo Scientific, USA). Gene expression was evaluated by quantitative polymerase chain reaction (qPCR) using the C1000 TouchTM thermal cycler and CFX96 Real-Time PCR Detection System (Bio-Rad, USA). The reactions were performed by using iTaq™ Universal SYBR® Green Supermix (Catalog number 1725120, Bio-Rad, USA) with immunomodulatory genes including indoleamine 2, 3-dioxygenase (IDO), hepatocyte growth factor (HGF), prostaglandin E2 (PGE-2), and interleukin-6 (IL-6), and the target genes of interest. The 18S gene, a housekeeping gene, served as the control. The primer sequences, based on the previous study, are displayed in S1 Table. The reaction conditions consisted of 40 cycles at a pre-denaturation temperature of 95˚C for 30 seconds, denaturation temperature of 95˚C for 5 seconds, and annealing/extension temperature of 60˚C for 30 seconds, respectively. The gene expression results were quantified by normalization using the $2^{-\Delta\Delta CT}$ method.

## Statistical analysis

The experiments were performed in triplicate, and the data were presented as mean ± standard deviation (SD). Data were analyzed using GraphPad Prism program version 9.2.0 (GraphPad Software, Inc., USA). Differences among groups were assessed by analysis of variance (ANOVA). Post-hoc multiple comparisons were made by the Tukey–Kramer test. Statistical significance was accepted at a p-value < 0.05.

## Results

### Isolation and characterization of canine adipose-derived stem cells

**Morphology.** After 2 days of isolation, the culture dishes were composed of several adherent cells with different morphology, including fibroblast-like or epithelial-like shapes. After 7

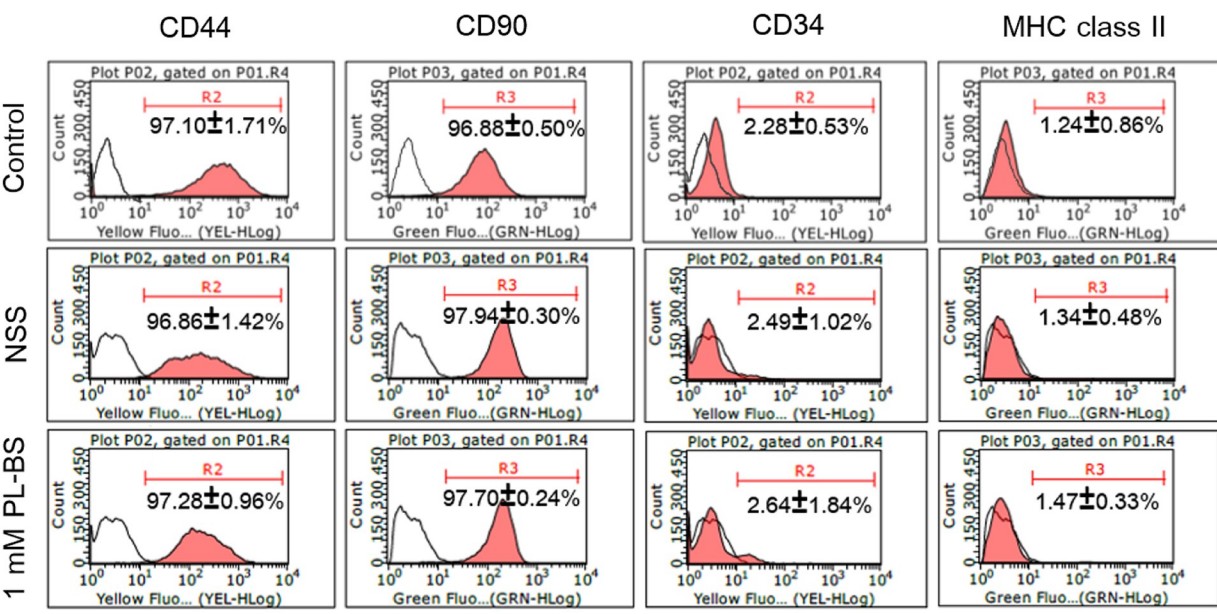

**Fig 1. Expression of specific surface antigens of canine adipose-derived mesenchymal stem cells in different conditions.** The stored cAD-MSCs highly expressed CD44 and CD90 but were deprived of CD34 and MHC class II, which was similar to the control group.

days, cells became more homogenous and the dominant cells were fibroblast-like. Cells reached 80–90% confluence at 14 days of isolation. In addition, the morphology of the three cell lines isolated from different donors was not notably different from one another (S1 Fig).

**Expression of specific surface antigens.** Based on flow cytometry, the isolated cells expressed CD44 ($97.10\pm1.71\%$), CD90 ($96.88\pm0.50\%$), CD34 ($2.28\pm0.53\%$), and MHC class II ($1.24\pm0.86\%$), as shown in Fig 1. In addition, a notable difference in expression of specific surface antigens was not observed among the three cell lines isolated from different donors (S2 Fig).

**Trilineage differentiation.** The isolated cells exhibited differentiation potential, which could be divided into adipogenic, chondrogenic, and osteogenic lineages, as illustrated in Fig 2. The red-colored fat droplets were seen in adipocytes. The chondrocytes were observed as blue-stained spheroidal cells. The calcium-deposited osteocytes were stained red. All three lines of cAD-MSCs exhibited osteogenic differentiation potential. In addition, the differences among three lines were not distinct (S3 Fig).

## Effect of proline-based solution (PL-BS) on cell viability, proliferation ability, mitochondrial membrane potential (MMP) and stemness of cAD-MSCs after storage in a hypothermic state

**The 1 mM PL-BS decreased cell death during hypothermic storage.** According to the flow cytometry results, the maintenance of cAD-MSCs under hypothermic conditions dramatically increased levels of apoptotic and necrotic cells in a time-dependent manner (Fig 3A and S4 Fig). However, storing cAD-MSCs in 1 mM PL-BS for 6 hours significantly improved cell viability by inhibiting early apoptosis and necrosis in comparison with other concentrations of both PL-BS and NSS, as shown in Fig 3A and 3B. Notably, only at 6 hours after storage was the number of live cells in 1 mM PL-BS ($90.71\pm1.20\%$) significantly higher than that in NSS ($77.19\pm2.19\%$, $p<0.01$). Besides, the cells in 1 mM PL-BS ($1.43\pm1.10\%$) had a significantly lower

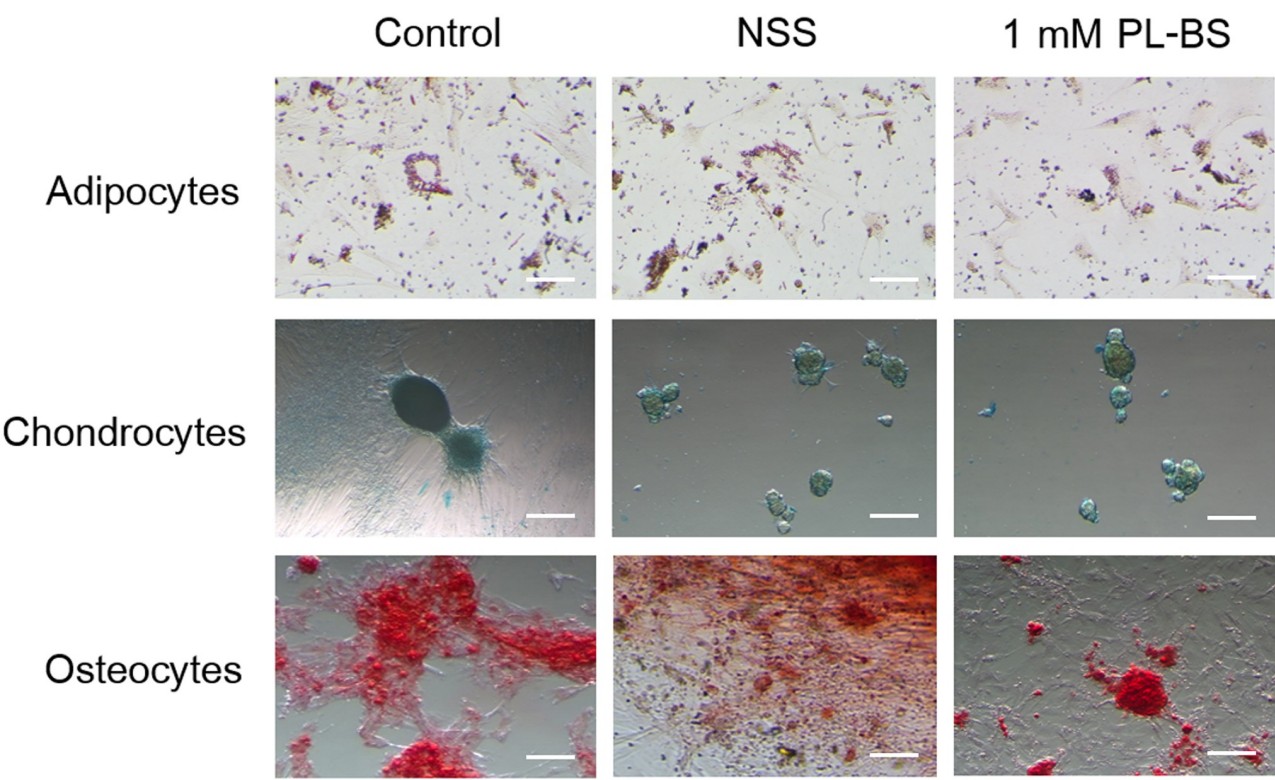

**Fig 2. The effect of a hypothermic solution on trilineage differentiation of canine adipose-derived mesenchymal stem cells.** The stored cAD-MSCs were able to differentiate into adipocytes, chondrocytes, and osteocytes. The red-stained fat droplets in adipocytes indicate adipogenic differentiation potential. The blue color indicates that chondrocytes are capable of producing proteoglycan in the extracellular matrix. The osteocytes that had accumulated calcium are shown in red color. Scale bars = 300 μm.

number of early apoptotic cells than those in NSS (9.91±4.97%, p<0.05), as displayed in Fig 3C.

**The 1 mM PL-BS preserved MMP during hypothermic storage.** MMP values were measured with a spectrophotometer, and are shown in Fig 4A. At 6 hours of storage, the cAD-MSCs stored in NSS (7236.25±552.61 au., p<0.0001) and 1 mM PL-BS (8243.50±290.11 au., p<0.001) showed a significant decrease in MMP compared to non-stored cells. However, the MMP of cAD-MSCs stored in 1 mM PL-BS was significantly higher than that of cAD-MSCs stored in NSS (p<0.05).

**Hypothermic storage decreased the proliferation ability of canine adipose-derived mesenchymal stem cells.** MTT assay revealed the proliferation rate of adherent cells, as shown in Fig 4B. At 6 hours of hypothermic storage, cAD-MSCs stored in NSS (0.3170±0.0095 au.) and 1 mM PL-BS (0.3288±0.0076 au.) showed a significant decrease in proliferation rate in comparison with that of non-stored cells (0.4616±0.0505 au., p<0.01). Moreover, no significant difference in proliferation rate was observed between storage in NSS and storage in 1 mM PL-BS.

**The hypothermic conditions did not negatively affect the expression of specific surface antigens.** An analysis using flow cytometry exhibited that there was no apparent difference in the expressions of CD44, CD90, CD34, and MHC class II between the control and cells stored in NSS and 1 mM PL-BS (Fig 1 and S2 Fig).

**The hypothermic solutions did not alter the differentiation potential of canine adipose-derived mesenchymal stem cells.** After fixing and staining differentiated cells with specific

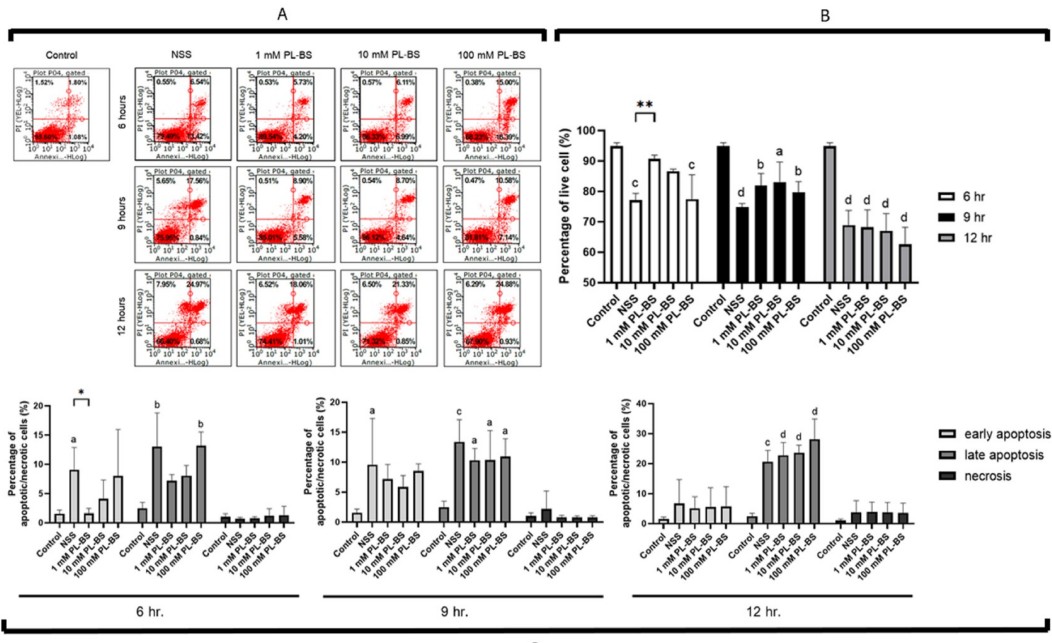

**Fig 3. The effect of a hypothermic solution on cell viability and apoptotic/necrotic fraction.** (A) The cAD-MSCs were stored in hypothermic solutions at concentrations of 1, 10, and 100 mM PL-BS or NSS for up to 6, 9, and 12 hours. (B) At 6 hours of hypothermic storage, the percentage of cAD-MSCs which had survived in 1 mM PL-BS was significantly higher than that in NSS (p<0.01). (C) The level of early apoptotic cells at 6 hours of hypothermic storage in 1 mM PL-BS was significantly lower than that in NSS (p<0.05). Remark symbols: Asterisk means values were significantly different from the NSS group; * = p<0.05 and ** = p<0.01. The lowercase letters a, b, c, and d above the bar indicate significant difference from the control group; a = p<0.05, b = p<0.01, c = p<0.001, and d = p<0.0001.

dyes, cAD-MSCs stored in NSS or 1 mM PL-BS maintained their trilineage differentiation ability, as shown in Fig 2. The differentiation abilities were confirmed by the positive staining of oil red O, alcian blue, and alizarin red S for adipogenic, chondrogenic and osteogenic differentiation, respectively. In addition, there was no obvious difference in differentiation ability among cAD-MSC lines derived from different donors (S3 Fig).

**The hypothermic solutions affected the expression of immunomodulatory genes in canine adipose-derived mesenchymal stem cells.** An analysis of gene expression with real-time RT-PCR indicated that the expressions of IDO and IL-6 were significantly upregulated in cAD-MSCs stored in NSS and 1 mM PL-BS, compared to non-stored cAD-MSCs (p<0.001), as demonstrated in Fig 5 and S2 Table. The expression level of HGF in cAD-MSCs stored in 1 mM PL-BS was significantly higher than that of cAD-MSCs stored in NSS and non-stored cells (p<0.05). In addition, the expression level of PGE-2 from cAD-MSCs stored in 1 mM PL-BS was lower than that of cAD-MSCs stored in NSS (p<0.05).

## Discussion

Mesenchymal stem cell therapy is promising for patients affected by several illnesses, such as degenerative, immune-mediated, and metabolic diseases [16]. The minimal criteria of MSC characteristics proposed by the International Society for Cell and Gene Therapy (ISCT) include (1) adherence to cell culture plates, (2) expression of specific surface antigens, such as CD73, CD90, and CD105, above 95% but expression of others, such as CD34, CD45, or HLA-II, lower than 2%, and (3) differentiation into three lineages, which are adipogenic,

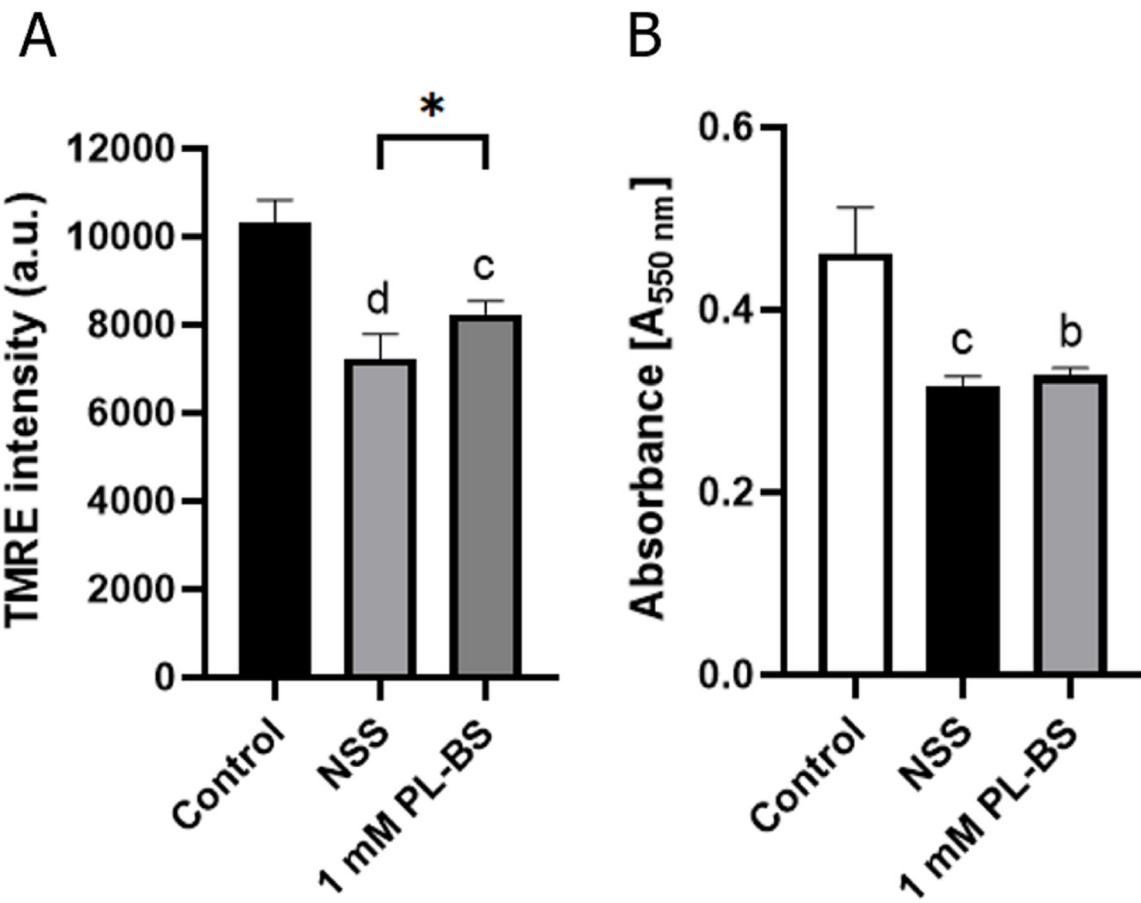

**Fig 4. Comparison of mitochondrial membrane potential and proliferation of canine adipose-derived mesenchymal stem cells in different conditions.** (A) Based on TMRE assay, the mitochondrial membrane potential (MMP) of cAD-MSCs stored in both NSS and 1 mM PL-BS for 6 hours, was significantly lower than that in the control group, (p<0.0001 and p<0.001, respectively). However, the MMP of cAD-MSCs stored in 1 mM PL-BS was significantly higher than that in NSS (p<0.05). (B) According to MTT assay, the proliferation rate of cAD-MSCs stored in NSS or 1 mM PL-BS was significantly lower than that in non-storage cells (p<0.01 and p<0.001, respectively). Remark symbols; Asterisk represents a significant difference (p<0.05). The lowercase letters b, c, and d indicate significant difference from the control group; b = p<0.01, c = p<0.001, and d = p<0.0001.

chondrogenic, and osteogenic lineage [17]. In veterinary medicine, one of the common sources for derivation of MSCs in dogs (Canidae) is adipose tissues. Therefore, cAD-MSCs have been widely studied both in basic and clinical research [5]. In the present study, newly derived cAD-MSCs exhibited fibroblast-like shapes, culture-plate-adhesion capacity, and trili-neage differentiation, which were similar to characteristics of human MSCs [14, 15, 18]. Besides, cAD-MSCs in our study had CD44 and CD90 expression of more than 95% but were deprived of CD34 and MHC class II expression, similar to a number of the previous reports [19–21].

According to the United States Food and Drug Administration (US FDA) recommenda-tions, the viability of stem cell products prior to clinical application should be greater than 70% [22]. Interestingly, approximately 70% of the stem cell clinical trials used NSS, which is approved by the US FDA as a cell-transportation solution [11, 23]. It is generally accepted and has been proven that prolonged storage of MSCs at 4°C, or under the physiological tempera-ture, provokes cell damage and death. Several studies have been performed on human MSCs

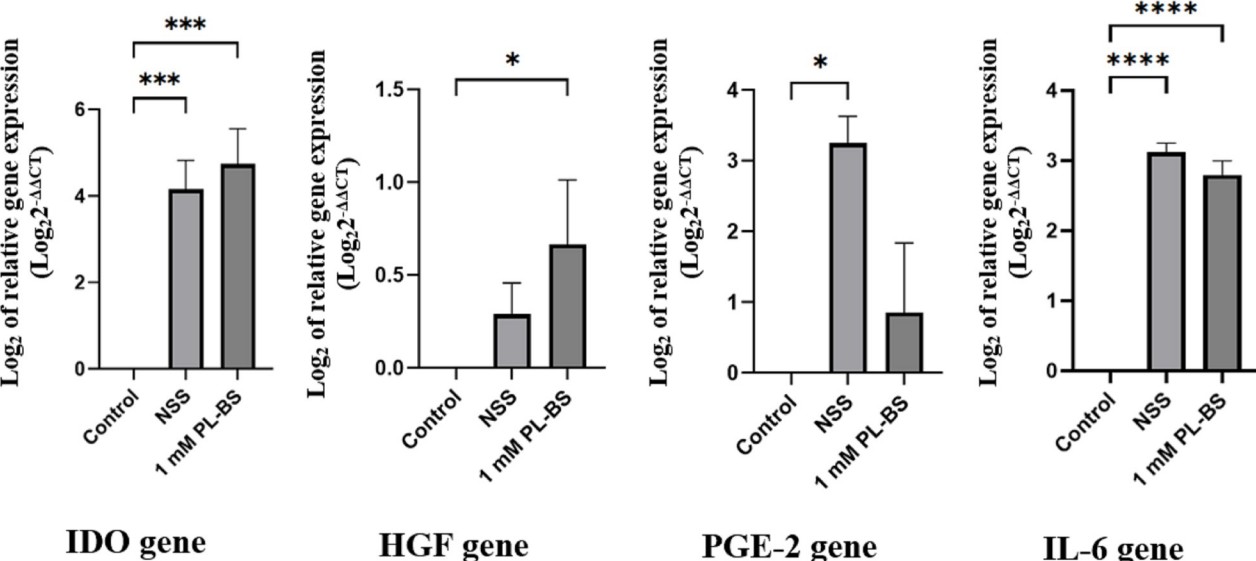

**Fig 5. Effect of a hypothermic solution on the expression of immunomodulatory genes.** Based on real-time RT-PCR, the level of IDO and IL-6 expression significantly increased when cAD-MSCs were stored in NSS or 1 mM PL-BS (p<0.001). The expression level of HGF significantly increased in cAD-MSCs stored in 1 mM PL-BS compared to non-stored cells (p<0.05). The PGE-2 expression level in NSS stored-cAD-MSCs was significantly higher than that in non-stored cells (p<0.05). Remark symbols; IDO = indoleamine 2, 3-dioxygenase, IL-6 = interleukin-6, HGF = hepatocyte growth factor, and PGE-2 = prostaglandin E2; Lowercase letters above the bars indicate a significant difference compared to the control group; a = p<0.05, c = p<0.001, and d = p<0.0001.

and contradictory results have been reported. For example, Pal and colleagues demonstrated that although the storage of human BMSCs in NSS at 4°C does not alter their differentiation capacity, the cells should be used within 6 hours in order to avoid cell death [24]. Chen and colleagues suggested that hUC-MSCs can be stored in NSS at 4°C for up to 6 hours without changing the expression of surface antigens, differentiation ability and immunomodulatory property of the cells. However, negative effects, such as a decrease in cell proliferation and adhesion capacity, could be consequences [12]. In contrast, the study of Sohn and colleagues demonstrated that NSS influences the differentiation ability and colony-forming units of hBMSCs in a time-dependent manner [13]. Nevertheless, the effect of NSS may vary depending on the source of the MSCs. Nofianti and colleagues' results demonstrated that hADSCs can maintain a cell viability of more than 70% after being stored in NSS for up to 48 hours; however, their proliferation rate significantly decreased after storage for 24 hours. The authors also suggested that hADSCs be applied for cell therapy within 24 hours [25]. Ra and colleagues reported that hADSCs can be stored in NSS up to 72 hours without changes in surface marker antigens [26]. Unfortunately, the results from Wu and colleagues showed that hADSCs cannot maintain a cell viability of over 70% in NSS at 4°C [27]. Although many studies have demonstrated that storing the cells in NSS at 4°C decreases cell viability in a time-dependent manner and varies according to the source of the cells, information regarding canine MSCs is still lacking. Therefore, NSS was selected as a basic solution for handling cAD-MSCs during storage in hypothermic conditions in the present study. The current study confirmed that NSS efficiently maintained cAD-MSC viability at 4°C for 6–9 hours. When the NSS was supplemented with 1 mM of proline, the early apoptotic rate of the cAD-MSCs decreased. This resulted in an improvement of cAD-MSC viability. The protective effect of proline on early apoptosis and cell viability was clearly observed after storing the cAD-MSCs at 4°C for 6 hours. From these results, it might be implied that 1 mM of proline improved the capacity of NSS for handling

cAD-MSCs under hypothermic conditions. Moreover, this solution did not affect expression of specific surface antigens and trilineage differentiation potential of cAD-MSCs. However, the proliferation ability of the stored cells was neither improved in NSS [12, 25] nor improved in 1 mM of supplemented proline, as demonstrated by the present study.

After being in hypothermic conditions, cells adapt themselves by reducing their metabolic rate due to decreased ATP production, oxygen demand, and energy storage degradation [28, 29]. Cell swelling can occur due to water retention resulting from the loss of $K^+$ and diffusion into the cells of $Na^+$ and $Cl^-$ [30]. Moreover, that swelling can progressively occur when cells are stored in a solution with abundant $Na^+$ and $Cl^-$. Furthermore, prolonged storage time might decrease mitochondrial membrane potential, which induces mitochondrial depolarization contributing to mitochondrial dysfunction and subsequently progressive ROS production or oxidative stress, which finally results in cell death [31–33].

Cryoprotective agents play a critical role in protecting the cells from freezing conditions (below 0°C). Besides, they can also protect the cells from cold temperatures (4°C) during cell transportation. Proline has been used as a cryoprotective agent for oocytes [34], human red blood cells [35], and human MSCs [36]. Proline is a natural non-toxic amino acid compound [37]. It has been identified as a penetrating cryoprotectant and acts as an osmoprotectant, helping balance osmotic pressure between cells and surrounding water during temperature stress or salt stress from $Na^+$, $K^+$, and $Cl^-$ [38]. In addition, proline stabilizes protein and macromolecules such as nucleic acid during cold stress [35, 39, 40]. Moreover, proline acts as an ROS scavenger during prolonged storage [41]. In the cells, proline metabolism helps increase glutathione production and maintain proper $NADP^+$/NADPH level, which assists energy homeostasis [39]. From the results of the present study, it could be implied that proline might maintain mitochondrial integrity during environmental stress and subsequently inhibit cell death [42].

In spite of supportive maintenance of cAD-MSC viability, PL-BS improved the immunomodulation of cAD-MSCs. The present study demonstrated that genes encoding IDO and IL-6 were significantly upregulated in cAD-MSCs stored in PL-BS. Normally, IDO is expressed during immunosuppressive processes via inhibition of pro-inflammatory T-cells as well as induction of the regulatory T-cell population [43]. IL-6 is a pro- and anti-inflammatory cytokine which plays a major role in MSC proliferation and the immunosuppressive capacity of hBMSCs [44, 45]. Moreover, IL-6 maintains the stemness of hBMSCs via ERK1/2-dependent mechanisms [46]. Thus, the immunosuppressive impact of cAD-MSCs can be improved by using PL-BS during hypothermic transportation. The present study showed that despite the upregulation of IDO and IL-6 genes, the level of HGF expression was still upregulated after storing cAD-MSCs in PL-BS. It has been demonstrated that HGF reduces inflammatory Th1 cells via dendritic cells and stimulates regulatory T cells to produce immunosuppressive cytokines which are beneficial in the treatment of autoimmune diseases [47]. Besides, the upregulation of HGF expression in cAD-MSCs in the present study might positively correlate with an improvement of cell viability as HGF plays vital roles in delaying senescence, enhancing osteogenic differentiation potential, and preserving mitochondrial function [48]. PGE-2 regulates macrophages and dendritic cells as well as T- and B-cells; this subsequently involves pro- and anti-inflammatory processes [49]. In addition, PGE-2 promotes the migration of MSCs [50] and the downregulation of PGE-2, resulting in decreased immunomodulatory properties [51]. In the present study, the storage of cAD-MSCs in PL-BS slightly upregulated the expression of PGE-2, but this upregulation was not as high as that from cAD-MSCs stored in NSS. This might be due to the presence of negative feedback mechanisms of the storage cells in order to decrease the excessive amount of proline by lowering PGE-2 production [52, 53]. Therefore,

the immunomodulatory capacity of cAD-MSCs can be improved by supplementation with TNF-α or IFN-γ during cultivation [54, 55].

## Conclusions

Proline-based solution had a short-term protective effect on cAD-MSCs stored in hypothermic conditions. The supplementation of proline in NSS decreased levels of early apoptotic cells, maybe by preserving mitochondrial function, together with improving the cell viability and stemness of cAD-MSCs. Therefore, proline-based solution can be applied for short-term hypothermic transportation and might be beneficial in clinical application.

## Supporting information

**S1 Fig. Morphology of isolated canine adipose-derived mesenchymal stem cells.** After 2 days of isolation, the adherent cells with different morphology, including fibroblast-like or epithelial-like shapes, were observed in the culture dishes. After 7 days, the cells had become more homogenous and those with a fibroblast-like shape had become dominant. The cells reached 80–90% confluence at 14 days of isolation. There was no notable difference in cell morphology among the three cell lines that had been isolated from different donors. All figures were performed at 10x magnification. Scale bars = 300 μm.
(TIF)

**S2 Fig. Expression of specific surface antigens of non-stored and stored canine adipose-derived mesenchymal stem cells was analyzed by flow cytometer.** The stored cells in NSS and 1 mM PL-BS highly expressed both CD44 and CD90 but were deprived of both CD34 and MHC class II. These results were not different from those of the control group.
(TIF)

**S3 Fig. The effect of a hypothermic solution on trilineage differentiation.** The post-stored cAD-MSCs exhibited differentiability into three lineages, including adipocytes, chondrocytes, and osteocytes. The fat droplets in adipocytes are stained red, indicating adipogenic differentiation potential. The blue color points out that chondrocytes are capable of producing proteoglycan in the extracellular matrix. The osteocytes with calcium accumulation are shown in red. Scale bars = 300 μm.
(TIF)

**S4 Fig. The effect of a hypothermic solution on cell viability and apoptotic/necrotic fraction.** The cAD-MSCs from two other donors decreased the number of live cells in a time-dependent manner during hypothermic conditions. However, storage of cAD-MSCs in 1 mM PL-BS for 6 hours considerably improved cell viability by decreasing early apoptosis and necrosis in comparison with storage in other concentrations of both PL-BS and NSS.
(TIF)

**S1 Table. Sequence of primers used for evaluating the expression of immunomodulatory genes.**
(DOCX)

**S2 Table. The effect of a hypothermic solution on the expression of immunomodulatory genes.** The gene expression results were quantified by normalization using the $2^{-\Delta\Delta CT}$ method and are shown as $Log_2$ values of relative gene expression.
(DOCX)

## Acknowledgments

The authors would like to thank the veterinarian and staff of iVET animal hospital for providing all the samples for the present study.

## Author Contributions

**Conceptualization:** Pongsatorn Horcharoensuk, Amarin Narkwichean, Ruttachuk Rungsiwiwut.

**Data curation:** Pongsatorn Horcharoensuk, Sunantha Yang-en, Ruttachuk Rungsiwiwut.

**Formal analysis:** Pongsatorn Horcharoensuk, Ruttachuk Rungsiwiwut.

**Investigation:** Pongsatorn Horcharoensuk, Ruttachuk Rungsiwiwut.

**Methodology:** Pongsatorn Horcharoensuk, Sunantha Yang-en, Ruttachuk Rungsiwiwut.

**Project administration:** Ruttachuk Rungsiwiwut.

**Supervision:** Amarin Narkwichean, Ruttachuk Rungsiwiwut.

**Validation:** Pongsatorn Horcharoensuk, Ruttachuk Rungsiwiwut.

**Visualization:** Ruttachuk Rungsiwiwut.

**Writing – original draft:** Pongsatorn Horcharoensuk, Sunantha Yang-en, Ruttachuk Rungsiwiwut.

**Writing – review & editing:** Pongsatorn Horcharoensuk, Amarin Narkwichean, Ruttachuk Rungsiwiwut.

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
