## [Decision Letter · Decision Letter 0]

15 Dec 2021

PONE-D-21-31968Proline-based solution maintains cell viability and stemness of canine adipose-derived mesenchymal stem cells after hypothermic storagePLOS ONE

Dear Dr. Rungsiwiwut,

Thank you for submitting your manuscript to PLOS ONE. After careful consideration, we feel that it has merit but does not fully meet PLOS ONE’s publication criteria as it currently stands. Therefore, we invite you to submit a revised version of the manuscript that addresses the points raised during the review process.

We look forward to receiving your revised manuscript.

Kind regards,

Nazmul Haque

Academic Editor

PLOS ONE

Whilst you may use any professional scientific editing service of your choice, PLOS has partnered with both American Journal Experts (AJE) and Editage to provide discounted services to PLOS authors. Both organizations have experience helping authors meet PLOS guidelines and can provide language editing, translation, manuscript formatting, and figure formatting to ensure your manuscript meets our submission guidelines. To take advantage of our partnership with AJE, visit the AJE website (http://aje.com/go/plos) for a 15% discount off AJE services. To take advantage of our partnership with Editage, visit the Editage website (www.editage.com) and enter referral code PLOSEDIT for a 15% discount off Editage services.  If the PLOS editorial team finds any language issues in text that either AJE or Editage has edited, the service provider will re-edit the text for free.

A clean copy of the edited manuscript (uploaded as the new *manuscript* file

“This research was funded by the Faculty of Medicine, Srinakharinwirot University (MED-GRAD-150; 511/2563 and MED-RES-200; 217/2563).”

“This research was funded by the Faculty of Medicine, Srinakharinwirot University (MED-GRAD-150; 511/2563 and MED-RES-200; 217/2563).”

We note that you have provided information within the Acknowledgements Section. Please note that funding information should not appear in the Acknowledgments section or other areas of your manuscript. We will only publish funding information present in the Funding Statement section of the online submission form.

“This research was funded by the Faculty of Medicine, Srinakharinwirot University (MED-GRAD-150; 511/2563 and MED-RES-200; 217/2563).”

5. Please upload a new copy of Figure 1A as the detail is not clear. Please follow the link for more information: https://blogs.plos.org/plos/2019/06/looking-good-tips-for-creating-your-plos-figures-graphics/

Reviewers' comments:

Reviewer's Responses to Questions

**Comments to the Author**

1. Is the manuscript technically sound, and do the data support the conclusions?

Reviewer #1: Partly

2. Has the statistical analysis been performed appropriately and rigorously? 

Reviewer #1: Yes

3. Have the authors made all data underlying the findings in their manuscript fully available?

Reviewer #1: No

4. Is the manuscript presented in an intelligible fashion and written in standard English?

Reviewer #1: No

5. Review Comments to the Author

Reviewer #1: The manuscript (PONE-D-21-31968_reviewer)

The manuscript demonstrates best conditions to store mesenchymal stem cells (MSCs) meant for clinical or translation application. The paper highlights proline ability to protect MSCs from apoptosis within 6hrs of hypothermic conditions. The proline protection extend to improve proliferation, and stemness ability of MSCs. The conclusion was that proline will be beneficial in case of short term transportation of MSCs.

Problems with the manuscript.

• Typos and errors are not acceptable such as “hu-man”, “col-leagues”, “sur-face”, “Live-” and others. Authors could benefit from professional editing services.

• For mitochondrial membrane potential assay authors should mention how many cells were loaded in 96 well-plate.

• For proliferation assay, authors should mention how many cells were loaded in 96 well-plate.

• Adding the figure legends between texts is confusing. Authors should add the legends at the end of manuscript after references.

• Authors should mention how many cells have been gated during flow cytometry experiments.

• Fig. 4 needs a scale bar to be added.

• Logical error when authors wrote, “when cAD-MSCs are stored in NSS and 1 mM PL-BS” It should be “when cAD-MSCs are stored in NSS or 1 mM PL-BS” since they are not combined together. Authors could use professional editing services.

• The authors claim that proline protected the mitochondria dysfunction. The present study did not support this notion. The sentence should be removed or modified into less affirm tone.

• The qPCR raw data need to be available as supplementary such as Fig.5.

• In line 494, authors should add “maybe” before “by preserving mitochondrial function”.

Minor issues

• Minor English punctuation errors.

I recommend major revision to the current version

6. PLOS authors have the option to publish the peer review history of their article (what does this mean?). If published, this will include your full peer review and any attached files.

Reviewer #1: **Yes: **Bahauddeen Alrfaei

---

## [Author Response · Author response to Decision Letter 0]

19 Jan 2022

Response to Editor and Journal requirements:

and 

Answer: We provided the manuscript according to PLOS ONE style requirements.

2. We suggest you thoroughly copyedit your manuscript for language usage, spelling,and grammar. If you do not know anyone who can help you do this, you may wish to consider employing a professional scientific editing service.

Whilst you may use any professional scientific editing service of your choice, PLOS has partnered with both American Journal Experts (AJE) and Editage to provide discounted services to PLOS authors. Both organizations have experience helping authors meet PLOS guidelines and can provide language editing, translation, manuscript formatting, and figure formatting to ensure your manuscript meets our submission guidelines. To take advantage of our partnership with AJE, visit the AJE website (http://aje.com/go/plos) for a 15% discount off AJE services. To take advantage of our partnership with Editage, visit the Editage website (www.editage.com) and enter referral code PLOSEDIT for a 15% discount off Editage services. If the PLOS editorial team finds any language issues in text that either AJE or Editage has edited, the service provider will re-edit the text for free.

Upon resubmission, please provide the following: The name of the colleague or the details of the professional service that edited your manuscript.

Answer: The language of our manuscript had been edited by the professional English-language editing staff at International College for Sustainability Studies of Srinakharinwirot University. The certificate of language editing is attached as a PDF.

Answer: Highlighting revised manuscript is provided and uploaded as a supporting information file.

A clean copy of the edited manuscript (uploaded as the new *manuscript* file

Answer: A clean edited manuscript is provided and uploaded as the new manuscript file.

“This research was funded by the Faculty of Medicine, Srinakharinwirot

University (MED-GRAD-150; 511/2563 and MED-RES-200; 217/2563).” Please state what role the funders took in the study. If the funders had no role, please state: " The funders had no role in study design, data collection and analysis, decision to publish, or preparation of the manuscript."

Answer: Due to the funders had no role in the present study, therefore, we state “The funders had no role in study design, data collection and analysis, decision to publish, or preparation of the manuscript”.

4. Thank you for stating the following in the Acknowledgments Section of your

manuscript: “This research was funded by the Faculty of Medicine, Srinakharinwirot University (MED-GRAD-150; 511/2563 and MED-RES-200; 217/2563).” We note that you have provided information within the Acknowledgements Section. Please note that funding information should not appear in the Acknowledgments section or other areas of your manuscript. We will only publish funding information present in the Funding Statement section of the online submission form. Please remove any funding-related text from the manuscript and let us know how you would like to update your Funding Statement. Currently, your Funding Statement reads

as follows: This research was funded by the Faculty of Medicine, Srinakharinwirot University (MED-GRAD-150; 511/2563 and MED-RES-200; 217/2563).

Answer: The funding statement has been removed from the Acknowledgment section.

5. Please upload a new copy of

Figure 1A as the detail is not clear. Please follow the link for more information:

https://blogs.plos.org/plos/2019/06/looking-good-tips-for-creating-your-plos-figures-graphics/

Answer: The resolution of Figure 1A has been improved.

6. In your Data Availability statement, you have not specified where the minimal data set underlying the results described in your manuscript can be found. PLOS defines a study's minimal data set as the underlying data used to reach the conclusions drawn in the manuscript and any additional data required to replicate the reported study findings in their entirety. All PLOS journals require that the minimal data set be made fully available. 

Answer: We uploaded the data set as the supporting information file (S2)

For more information about our data policy, please see http://journals.plos.org/plosone/s/data-availability.

Upon re-submitting your revised manuscript, please upload your studys minimal underlying data set as either Supporting Information files or to a stable, public repository and include the relevant URLs, DOIs, or accession numbers within your revised cover letter. 

For a list of acceptable repositories, please see

http://journals.plos.org/plosone/s/data-availability#loc-recommended-repositories.

Any potentially identifying patient information must be fully anonymized.

Important: If there are ethical or legal restrictions to sharing your data publicly, please explain these restrictions in detail. Please see our guidelines for more information on what we consider unacceptable restrictions to publicly sharing data:

http://journals.plos.org/plosone/s/data-availability#loc-unacceptable-data-access-restrictions.

Note that it is not acceptable for the authors to be the sole named individuals responsible for ensuring data access.

Reviewer #1: The manuscript (PONE-D-21-31968_reviewer)

The manuscript demonstrates best conditions to store mesenchymal stem cells (MSCs) meant for clinical or translation application. The paper highlights proline ability to protect MSCs from apoptosis within 6hrs of hypothermic conditions. The proline protection extends to improve proliferation, and stemness ability of MSCs. The conclusion was that proline will be beneficial in case of short-term transportation of MSCs.

Problems with the manuscript.

Typos and errors are not acceptable such as “hu-man”, “col-leagues”, “sur-face”, “Live-” and others. Authors could benefit from professional editing services.

• Answer: All typos and errors have been corrected in the revised manuscript by professional grammar editing services from the International College for Sustainability Studies, Srinakharinwirot University. 

For mitochondrial membrane potential assay authors should mention how many cells were loaded in 96 well-plates.

• Answer: We added the cell number (1x105 cells/well) that was used in the mitochondrial membrane potential assay as the reviewer suggested. The cell number appear in line 168 of the revised manuscript.

For proliferation assay, authors should mention how many cells were loaded in 96 well-plates.

• Answer: We added the cell number (1x104 cells/well) that was used in the proliferation assay as the reviewer suggested. The cell number appear in line 175 of the revised manuscript.

Adding the figure legends between texts is confusing. Authors should add the legends at the end of manuscript after references.

• Answer: We moved all of the figure legends to the end of the revised manuscript (lines 639–715) as the reviewer suggested. 

Authors should mention how many cells have been gated during flow cytometry experiments.

• Answer: The number of cells that were used in the flow cytometry assay was added into both the cell viability and the apoptotic/necrotic fraction assay and the expression of specific surface antigens assay, which appeared in lines 163 and 191 of the revised manuscript.

Fig. 4 needs a scale bar to be added.

• Answer: The scale bars were embedded in Fig 4 as the reviewer suggested.

Logical error when authors wrote, “when cAD-MSCs are stored in NSS and 1 mM PL-BS” It should be “when cAD-MSCs are stored in NSS or 1 mM PL-BS” since they are not combined together. Authors could use professional editing services.

• Answer: We amended the texts into “when cAD-MSCs are stored in NSS or 1 mM PL-BS” which is shown in line 675 of the revised manuscript.

The authors claim that proline protected the mitochondria dysfunction. The present study did not support this notion. The sentence should be removed or modified into less affirm tone.

• Answer: We amended the sentence “proline protected the mitochondria dysfunction” to “proline might maintain mitochondrial integrity” which is shown in line 394-395 of the revised manuscript.

The qPCR raw data need to be available as supplementary such as Fig.5.

• Answer: We added the qPCR data in supplementation S2 Table as the reviewer suggested, which appears in lines 712-715.

In line 494, authors should add “maybe” before “by preserving mitochondrial function”.

• Answer: The word "maybe" was added to the sentence before "by preserving mitochondrial function" as the reviewer suggested, and it appears in line 427 of the revised manuscript.

Minor issues

Minor English punctuation errors.

• Answer: The revised manuscript has undergone grammatical correction by professional grammar editing services from International College for Sustainability Studies, Srinakharinwirot University.

---

## [Decision Letter · Decision Letter 1]

17 Feb 2022

Proline-based solution maintains cell viability and stemness of canine adipose-derived mesenchymal stem cells after hypothermic storage

PONE-D-21-31968R1

Dear Dr. Rungsiwiwut,

We’re pleased to inform you that your manuscript has been judged scientifically suitable for publication and will be formally accepted for publication once it meets all outstanding technical requirements.

Kind regards,

Nazmul Haque

Academic Editor

PLOS ONE

Additional Editor Comments (optional):

Reviewers' comments:

Reviewer's Responses to Questions

**Comments to the Author**

1. If the authors have adequately addressed your comments raised in a previous round of review and you feel that this manuscript is now acceptable for publication, you may indicate that here to bypass the “Comments to the Author” section, enter your conflict of interest statement in the “Confidential to Editor” section, and submit your "Accept" recommendation.

Reviewer #1: All comments have been addressed

2. Is the manuscript technically sound, and do the data support the conclusions?

Reviewer #1: Yes

3. Has the statistical analysis been performed appropriately and rigorously? 

Reviewer #1: Yes

4. Have the authors made all data underlying the findings in their manuscript fully available?

Reviewer #1: Yes

5. Is the manuscript presented in an intelligible fashion and written in standard English?

Reviewer #1: Yes

6. Review Comments to the Author

Reviewer #1: (No Response)

7. PLOS authors have the option to publish the peer review history of their article (what does this mean?). If published, this will include your full peer review and any attached files.

Reviewer #1: **Yes: **Bahauddeen Alrfaei

---

## [Editor Report · Acceptance letter]

21 Feb 2022

PONE-D-21-31968R1 

Proline-based solution maintains cell viability and stemness of canine adipose-derived mesenchymal stem cells after hypothermic storage 

Dear Dr. Rungsiwiwut:

I'm pleased to inform you that your manuscript has been deemed suitable for publication in PLOS ONE. Congratulations! Your manuscript is now with our production department. 

Kind regards, 

on behalf of

Dr. Nazmul Haque 

Academic Editor

PLOS ONE